# GRAPE: GRAPH REASONING WITH ANONYMOUS PATH ENCODERS

## ABSTRACT

Since the introduction of retrieval-augmented generation (RAG), a standard component of large language model (LLM) reasoning pipelines has been the navigation of a knowledge base (KB) to generate answers grounded in retrieved sources. However, recent studies show that LLMs struggle with complex queries requiring deep reasoning and interdependent knowledge, often leading to hallucinations. While several methods have been proposed to mitigate this issue, most rely on multiple additional calls to the LLM to decompose and validate reasoning steps, thereby increasing inference cost and latency. In this paper, we introduce GRAPE (Graph Reasoning with Anonymous Path Encoders), a framework that leverages path encodings over uncertain nodes and relations in knowledge graphs (KGs) to heuristically guide KB navigation. Rather than depending on a fully LLM-native retrieval pipeline, GRAPE replaces repeated model calls with encoder-only models that act as a semantic fuzzy query-matching engine. Experiments across multiple multi-hop QA benchmarks show that GRAPE achieves up to 85% faster inference than LLM-based pipelines, while consistently matching or exceeding state-of-the-art accuracy. These results demonstrate that encoder-only hybrid reasoning pipelines provide a practical and scalable alternative to expensive LLM-native retrieval, combining efficiency, robustness, and strong generalization.

## 1 INTRODUCTION

Writing a question is easy for humans, but despite strong performance across many tasks, large language models (LLMs) still struggle to answer queries that demand complex reasoning or deep, long-tail knowledge Petroni et al. (2021); Talmor et al. (2019); Malek et al. (2025); Tan et al. (2025b). Modern systems address this by supplying LLMs with retrieval mechanisms that surface evidence at inference time, mitigating the limits of pretraining Wei et al. (2022); Yao et al. (2023a); Besta et al. (2024); Jeong et al. (2024). Recently, structural signals from *knowledge graphs* (KGs) have been integrated for knowledge-graph question answering (KGQA), where answers are grounded in graph topology Sun et al. (2023); Li et al. (2023b); Agrawal et al. (2024). Since the emergence of retrieval-augmented generation (RAG) Lewis et al. (2020), many pipelines let the LLM act as an "agent," issuing pre-formatted prompts to query, filter, and reason over the KG Sun et al. (2023); Tan et al. (2025a); Luo et al. (2024); Walter & Bast (2025).

This design has two drawbacks. **(i) Computational/latency:** multi-hop reasoning requires traversing and exploring the graph; adding LLM latency to every step makes end-to-end inference impractical at scale Oche et al. (2025). **(ii) Cost:** LLMs are flexible but expensive; driving the entire retrieval loop with recurrent model calls quickly becomes prohibitive Chen et al. (2023).

To address these issues, we propose GRAPE—a graph reasoning system with an *anonymous paths encoder*. GRAPE outsources LLM usage from the exploration loop to encoders that perform *fuzzy* path matching between a query-derived anonymous KG pattern and the target KG, where nodes may be variables to be resolved. In doing so, we reformulate KG retrieval as encoder-guided graph matching, preserving reasoning structure while reducing LLM calls, latency, and cost, and enabling scalable KGQA.

## 2 RELATED WORK

In general, LLM-based question answering is highly susceptible to hallucination, where plausible but unfounded intermediate steps lead to errors in complex reasoning tasks Sadat et al. (2023); Huang et al. (2025); Jiang et al. (2024). Early efforts to mitigate this problem emerged with Chain-of-Thought (CoT) Wei et al. (2022) and its many variants—such as Auto-CoT Zhang et al. (2022b), Complex-CoT Fu et al. (2023), Zero-Shot CoT Kojima et al. (2022), Tree-of-Thought (ToT) Yao et al. (2023a), and Graph-of-Thought (GoT)Besta et al. (2024)—which introduce intermediate reasoning steps into the prompt to guide the model's reasoning and reduce hallucination. While these approaches produce more grounded responses, their effectiveness on knowledge-intensive or multi-hop reasoning tasks remains constrained by the model's internal training data, motivating the integration of external knowledge sources during reasoning Luo et al. (2024); Li et al. (2023b); Yao et al. (2023b).

One natural extension has been to use knowledge graphs (KGs) as an external interface that models can query during reasoning, thereby alleviating both the training-data limitation and hallucination Pan et al. (2024); Agrawal et al. (2024). Early knowledge-graph question answering (KGQA) approaches attempted to embed KG structure directly into transformer architectures at training and fine-tuning time Zhang et al. (2022a); Peters et al. (2019); Li et al. (2023b). However, this strategy sacrifices scalability, as the model becomes tightly coupled to the training-time KG, and reduces representational reasoning flexibility Wen et al. (2024); Hu et al. (2024). More recent KGQA systems instead introduce additional extraction and processing steps within the native LLM-based QA pipeline, which circumvents the scalability issue Ma et al. (2024); Jin et al. (2024).

Among this new family of models, the most relevant to our work are: *Think on Graph* (ToG) Sun et al. (2023), which performs beam search over KG paths to identify the most likely reasoning chains; *Think on Graph 2.0* (ToG-2) Ma et al. (2024), which extends ToG by tightly coupling KG traversal with entity-linked document retrieval, allowing structured and unstructured sources to reinforce each other in multi-hop reasoning; and *Paths on Graphs* (PoG) Tan et al. (2025a), which adds a pruning step using an off-the-shelf model such as SBERT Reimers & Gurevych (2019) to filter candidate paths after the expansion and search phases, thereby reducing the number of reasoning calls.

Although these approaches achieve significantly better performance on complex reasoning tasks than CoT-based methods, they incur substantial overhead in both cost and latency, as their retrieval pipelines require repeated LLM calls Chen et al. (2023); Oche et al. (2025); Wang et al. (2024); **?**. The bottleneck typically lies in the graph exploration phase, where most methods rely on iterative expansion, pruning, and evaluation, invoking the LLM at every step of the planned reasoning process during the initial query resolution Su et al. (2025). Some recent work has explored dynamic graph pruning for KGQA—for example, KAPING Baek et al. (2023), which prunes at the triplet level (entity–relation–entity) using semantic similarity between questions and triplets; KG-GPT Kim et al. (2023), which decomposes the main question into sub-questions to retrieve relevant relations; and PipeNet Su et al. (2024), which prunes nodes based on their distance to query-matched sub-spans. While these methods improve efficiency, their pruning operates locally at the triplet level, often overlooking the global graph structure and sacrificing performance on long-range dependencies Tan et al. (2025a).

To the best of our knowledge, no existing KGQA method dynamically prunes graph exploration paths while considering the global graph structure and avoiding additional LLM calls. To this end, we introduce GRAPE, a simple framework based on path encoding with uncertain (anonymous) entities that reformulates KGQA as a fuzzy graph matching problem between the query graph and the KG.

## 3 METHODOLOGY

The GRAPE framework aims to reduce reliance on repeated external calls to the LLM within the retrieval pipeline by replacing LLM-based search with fuzzy path matching between the knowledge graph (KG) and an Anonymous Knowledge Graph (AKG) constructed from the query. Matching

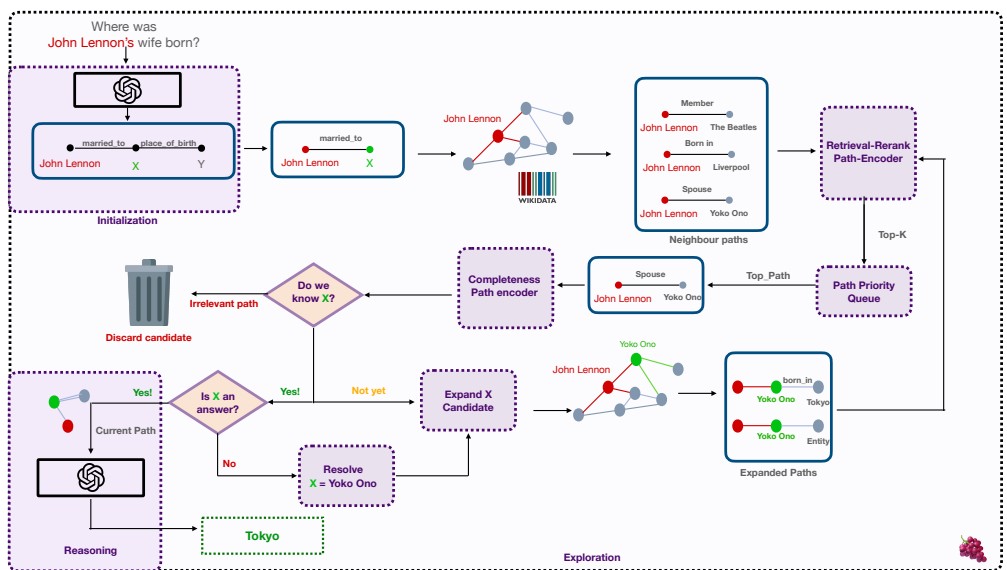

Figure 1: **GRAPE pipeline.** (1) *Initialization*: convert the query to an Anonymous Knowledge Graph (AKG) and link known entities. (2) *Exploration*: encoder-guided fuzzy path matching with relation clustering, a fixed-size beam, and completeness labels $\{S, C, D\}$ (Stop/Continue/Drop). (3) *Reasoning*: a single LLM call aggregates top candidates and their supporting KG paths into final answers.

proceeds by iteratively disambiguating the unknown nodes in the AKG, evaluating path completeness through a dedicated encoder, and pruning candidates via top-N semantic similarity. Once the anonymous entities have been resolved, a single final call to the LLM is used to generate the answer. In line with prior work, the framework is organized into three components: initialization, exploration, and reasoning.

## 3.1 FORMAL SETTING OF KGQA

Let $\mathcal{G} := (\mathcal{E}, \mathcal{R}, \mathcal{T})$ denote a knowledge graph, where $\mathcal{E}$ is the set of entities (nodes), $\mathcal{R}$ the set of relations (edges), and $\mathcal{T} \subseteq \mathcal{E} \times \mathcal{R} \times \mathcal{E}$ the set of triplets defining the graph in the form (head, relation, tail). A path in $\mathcal{G}$ is defined as

$$P := [(e_0, r_0, e_1), (e_1, r_1, e_2), \ldots, (e_{k-1}, r_k, e_k)] \in \mathcal{P}(\mathcal{G}),$$

where $\mathcal{P}(\mathcal{G})$ denotes the set of all valid paths in the graph.

Formally, a system for knowledge graph question answering is a mapping from a query in natural language to a subset of supported evidence (i.e., an answer set):

$$f : \mathcal{L} \times \mathcal{G} \longrightarrow \mathcal{P}(\mathcal{E})$$

$$(q, \mathcal{G}) \longmapsto \mathcal{A}_q := \big\{ e \in \mathcal{E} \mid \exists P = [(e_0, r_1, e_1), \ldots, (e_{k-1}, r_k, e_k)] \in \mathcal{P}(\mathcal{G}),$$
$$P \models q \ \wedge \ e = e_k \big\},$$

where $\mathcal{L} := \langle \mathbb{V} \rangle$ is the language set generated from a vocabulary $\mathbb{V}$ and the relation $A \models B$ indicates that the structure $A$ satisfies the conditions expressed by $B$.

Modern LLM-based systems often decompose the initial query $q$ into a sequence of subqueries (a resolution plan) $(q_0, \ldots, q_N)$. The resolution process then follows the recursion

$$f_k = \begin{cases} f(q_0, \mathcal{G}; \varnothing) & = \mathcal{A}_0, \\ f(q_k, \mathcal{G}; f_{k-1}) & = \mathcal{A}_k, \quad k \geq 1, \end{cases}$$

where $f(\cdot, \cdot; H)$ denotes a call to the system with additional context $H$ (in this case, the previous answer set), and the recursion terminates with $f_k = \mathcal{A}_q$.

## 3.2 INITIALIZATION

Using an LLM to extract a resolution plan is an easy and flexible way to determine an action plan from the initially given query. However, this makes a retrieval model entirely dependent on the LLM during the exploration process, as decomposing the query necessarily produces sub-queries that lose the global structure of the original intent. This, in turn, requires repeated calls to the LLM to maintain coherence, which increases inference cost and introduces additional opportunities for hallucination.

By contrast, our approach reformulates the query as an *Anonymous Knowledge Graph (AKG)*, pre-serving its overall structure while allowing unknown entities to remain as variables to be resolved (see Figure 1). For example, the query *"Director of a movie in which Tom Hanks acted"* is translated into the AKG

$$\text{Tom Hanks} \xrightarrow{\text{acted\_in}} X_1 \xrightarrow{\text{directed\_by}} X_2,$$

where $X_1$ denotes an anonymous variable corresponding to an unknown movie and $X_2$ an anonymous variable representing the director to be returned as the answer.

Formally, for a given query $q$ we define its associated *Anonymous Knowledge Graph* (AKG) as

$$\hat{\mathcal{G}}_q = \big(E_q, R_q, T_q\big),$$

where

$$E_q = E \cup \hat{E}, \qquad T_q = T \cup \hat{T}_1 \cup \hat{T}_2.$$

Here $E$ denotes the set of entities explicitly mentioned in the query, $\hat{E}$ the set of anonymous variables introduced during query parsing, and $R_q$ the set of relations extracted from $q$. The set $T$ contains triplets composed only of known entities, $\hat{T}_1$ the triplets involving one anonymous variable, and $\hat{T}_2$ the triplets involving two anonymous variables.

Instead of LLM planned query decomposition, the resolution process can be expressed as

$$\hat{\mathcal{G}}_q^{(k)} = \begin{cases} \hat{\mathcal{G}}_q, & k = 0, \\ \text{resolve}\big(\hat{\mathcal{G}}_q^{(k-1)}, X_i \mapsto e^*\big), & k \geq 1, \end{cases}$$

$$e^* = \arg\max_{e \in \mathcal{E}} \ s\big(X_i \mid \hat{\mathcal{G}}_q^{(k-1)}, e, \mathcal{G}\big), \tag{1}$$

where at each step a single anonymous variable $X_i \in \hat{E}$ is substituted with a candidate entity $e^* \in \mathcal{E}$ from the knowledge graph $\mathcal{G}$. The scoring function $s(\cdot)$ evaluates the compatibility of $e$ with the current state of the AKG $\hat{\mathcal{G}}_q^{(k-1)}$ and the evidence in $\mathcal{G}$. The operator $\text{resolve}(\hat{\mathcal{G}}_q^{(k-1)}, X_i \mapsto e^*)$ then updates the graph by replacing $X_i$ with $e^*$ in all affected triplets, yielding the new state $\hat{\mathcal{G}}_q^{(k)}$. The process continues until all anonymous variables are resolved, resulting in the fully instantiated query graph $\hat{\mathcal{G}}_q^{(*)}$.

In practice, we generate the initial AKG graph $\mathcal{G}_q$ by using a few shots example query to an LLM (in our case GPT-4o OpenAI et al. (2024)) that can be found in Section A.1. We also follow ToG Sun et al. (2023) technique to map all non-anonymous entities to node candidates in the KG.

## 3.3 EXPLORATION

Exploration is framed as a graph matching problem between the AKG $\hat{\mathcal{G}}_q$ and the KG $\mathcal{G}$. At each iteration, we attempt to resolve one anonymous variable $X \in \hat{E}$ by aligning the set of AKG triplets that involve it,

$$T_X := \big\{\, (h, r, t) \in T^{(k)} \ \mid \ h = X \ \vee \ t = X \,\big\} \subseteq T^{(k)},$$

with candidate continuations drawn from $\mathcal{G}$. Unlike exact matching, the process is *fuzzy*: a relation $r \in R_q$ in the AKG does not need to correspond to an identical edge in $\mathcal{G}$, but can be approximated by semantically similar relations or by a short path.

Formally, let the query graph at iteration $k$ be

$$\hat{\mathcal{G}}_q^{(k)} = \left( E \cup \hat{E}, \ R_q, \ T^{(k)} \right),$$

where $E$ are bound entities, $\hat{E}$ are still-unresolved variables, and $T^{(k)}$ is the current set of triplets (some involving anonymous nodes). We maintain a beam $\mathcal{F}_k$ of partial paths with fixed beam size $b$ and a maximum expansion depth $D_{\max}$ (in our experiments $b = 10$, $D_{\max} = 3$) as possible candidates to align $T_X$. The exploration process can then be decomposed into the following phases:

**Candidate generation.**   Given the frontier entity $e_m$ of a path $P \in F_k$ in the beam, we enumerate its one-hop neighborhood $\mathcal{N}(e_m) = \{(e_m, r, v)\} \in \mathcal{T}$. Each neighbor induces a candidate continuation $P' = P \oplus (e_m, r, v)$. We will expand the relations for every path in the beam sequentially and according to its similarity score.

**Scoring.**   Each candidate extension $P'$ is scored with two components:

1. A *retrieve–rerank* score $s^{\mathrm{rank}}(T_{X_k}, q, (e_m, r, v) \mid P)$, computed in two stages: first, a bi-encoder retriever (paraphrase-MiniLM-L3-v2 Reimers & Gurevych (2019)) evaluates the semantic similarity between the query and a plain-text representation of the path $P' = P \bigoplus (e_m, r, v)$; second, a cross-encoder reranker (fine-tuned from bge-reranker-v2-m3 Chen et al. (2024)) compares the local neighborhood of the anonymous entity $X_k$ (i.e., its incident triplets $T_{X_k}$) with the candidate continuation $P'$ and outputs a score in [0,1] depending on semantic similarity of both sets.

2. A *completeness score* $h_\theta(T_{X_k}, P') \in \{S, C, D\}$, predicted by a multiclass classifier fine-tuned from DeBERTa-v3-Large He et al. (2021), where $S$ denotes *Stop* (path already satisfies $T_{X_k}$), $C$ denotes *Continue* (expand further), and $D$ denotes *Drop* (irrelevant). Only paths predicted as $S$ or $C$ are retained.

Therefore, the scoring function in eq. (1) can be instantiated as:

$$s\left( X \mid \hat{\mathcal{G}}_q^{(k)}, e_m, \mathcal{G} \right) = \mathbb{1}[h_\theta(T_{X_k}, P') = S] \ + \ \mathbb{1}[h_\theta(T_{X_k}, P') = C] \, s^{\mathrm{rank}}\!\left( T_{X_k}, q, (e_m, r, v) \mid P \right)$$

Where $\mathbb{1}(\cdot)$ denotes the indicator function. A detailed explanation of the training of these models can be found in Section A.2.

**Clustering and representatives.**   To control high fan-out, expanded neighbors $\mathcal{N}(e_m)$ are clustered by relation, $\mathcal{N}(e_m) = \bigcup_r \mathcal{C}_r$, with $\mathcal{C}_r = \{(e_m, r, v) : v \in \mathcal{N}(e_m)\}$. From each cluster, we select a representative

$$v_r^* = \arg \max_{(r,v) \in \mathcal{C}_r} s^{\mathrm{rank}}(q, (e_m, r, v) \mid P),$$

forming the representative set $\mathcal{R}(e_m) = \{(e_m, r, v_r^*)\}_r$. Clusters are only decompressed when selected, which reduces redundant scoring.

**Beam update and backtracking.**   During disambiguation of $T_{X_k}$, after generating and scoring new candidates, we keep the beam size fixed at $b$ by retaining the top-$b$ paths and discarding the rest. The next path to expand is the highest-scoring one in the beam. If further expansion of that path becomes unpromising (its score degrades), we continue with the next best path in the current beam, allowing the search to switch branches without additional machinery.

**Termination.**   A path stops when $h_\theta(q, P) = \mathrm{S}$ or $\mathrm{depth}(P) = D_{\max}$. The surviving paths $\mathcal{F}_K$ define the candidate answer set by their terminal entities.

| Method | WebQSP | Adv HotpotQA | QALD-10-en | FEVER | CREAK | Zero-Shot RE | CWQ | Simple Questions | Web Questions | T-REx |
|---|---|---|---|---|---|---|---|---|---|---|
| ToG-2 (GPT-3.5) | 81.1 | 42.9 | 54.1 | 63.1[†] | 93.5 | 91.0 | – | – | – | – |
| ToG (GPT-4) | 76.2 | 26.3 | 50.2 | 52.7 | **93.8** | 88.0 | 57.1 | 53.6 | 54.5 | 76.8 |
| PoG-E (GPT-4) | 95.4 | – | – | – | – | – | 78.5 | 81.2 | 82.0 | – |
| PoG (GPT-4) | **96.7** | – | – | – | – | – | 81.4 | **84.0** | 84.6 | – |
| IO prompt (ChatGPT) | 63.3 | – | 42.0 | – | 89.7 | 27.7 | 37.6 | 20.0 | 48.7 | 33.6 |
| CoT (ChatGPT) | 62.2 | – | 42.9 | – | 90.1 | 28.8 | 38.8 | 20.3 | 48.5 | 32.0 |
| CoK (6-shot) | – | 35.4 | – | 58.5 | – | – | – | – | – | – |
| RoG | 85.7 | – | – | – | – | – | 62.6 | – | – | – |
| RRKG | 91.5 | – | – | – | – | – | 68.7 | – | – | – |
| **GRAPE (ours)** | 94.74 ± 3.87 | **60.31** ± 1.39 | **65.67** ± 3.84 | **79.12** ± 4.04 | 92.77 ± 3.74 | **94.20** ± 2.79 | 80.15 ± 2.17 | 82.87 ± 2.07 | **88.15** ± 4.87 | **94.86** ± 3.57 |

Table 1: Best per column in **bold**. Metrics: Hits@1 for WebQSP, AdvHotpotQA, QALD-10-en, Zero-Shot RE, CWQ, SimpleQuestions, WebQuestions, T-REx; Accuracy for FEVER and CREAK. GRAPE results are mean ± std over 5 runs. [†] FEVER for ToG-2 reported with 3-shots.

### 3.4 REASONING

We perform a single aggregation step with an LLM GPT-4o; prompt in the Section A.1). The top candidate entities and their supporting KG paths are provided to the model together with the original query and the AKG. The LLM selects the best answer(s) when multiple candidates remain (i.e., no path has been labeled $S$ by the completeness classifier).

Because we retain the KG relations used to reach each disambiguated entity (the logical path schema), we subsequently issue a SQL query over the KG store to retrieve any additional entities that instantiate the same path pattern, thereby covering multi-answer queries. The final answer set is the union of the LLM selection and the matched entities, deduplicated.

## 4 EXPERIMENTS

### 4.1 EXPERIMENTAL DESIGN

Following prior work Sun et al. (2023); Tan et al. (2025a); Ma et al. (2024); Li et al. (2023b), we evaluate GRAPE on ten datasets spanning KGQA, open-domain QA, slot filling, and fact checking. For KGQA we use four multi-hop sets—CWQ Talmor & Berant (2018), WebQSP Yih et al. (2016), QALD-10 Perevalov et al. (2022), and AdvHotpotQA Pan et al. (2024)—and one single-hop set, SimpleQA Bordes et al. (2015). All KGQA experiments are executed over the Wikidata Vrandečić & Krötzsch (2014) knowledge graph. For open-domain QA we use WebQuestions Berant et al. (2013) for slot filling, T-REx Elsahar et al. (2018) and ZeroShotRE Petroni et al. (2021); and for fact checking, FEVER Thorne et al. (2018) and CREAK Onoe et al. (2021). Our primary metric is Hits@1 Li et al. (2023a); Baek et al. (2023); Sun et al. (2023). We also report average LLM calls and wall-clock time per query. Because LLMs can be non-deterministic even at temperature 0.0 Lops et al. (2024); Atil et al. (2025), we run each method five times over the full benchmark (over one month) and report mean ± standard deviation. Ablations examine (i) incremental addition of retrieval components starting from a vanilla LLM-only retriever, (ii) the effect of beam width and depth, and (iii) different LLM backbones and the completeness cross-encoder. Base configuration details are provided in the Section A.4.

### 4.2 MAIN RESULTS

### 4.3 COMPARSION WITH PREVIOUS WORK

In this experiment we compare Hits@1 for question–answer datasets and accuracy for claim–verification datasets. In Table 1 we can observe that GRAPE pushes performance on slot-filling benchmarks: on T-REx we obtain 94.86, clearly above all baselines in the table, and on Zero-Shot RE 94.20, slightly surpassing ToG-2 (91.0). Because relations in these datasets are generally shallow by construction, they rarely require deep reasoning; in practice, GRAPE's completeness mechanism tends to stop early (often at one hop), keeping the reasoning burden on the LLM small (see Appendix). By contrast, AdvHotpotQA and QALD-10-en are more knowledge-

| System | LLM calls (avg.) | Exec. time / query (avg., s) |
|---|---|---|
| ToG | 16.3 | 69.3 |
| ToG-2 | 5.4 | 27.3 |
| PoG | 7.8 | 34.6 |
| CoK | 11.0 | 30.1 |
| **GRAPE (ours)** | **2.4** | **10.4** |

Table 2: Average LLM calls and average execution time per query. Lower is better.

intensive and frequently need multi-hop evidence. Here, GRAPE significantly outperforms alternative approaches—60.31 vs 42.9 on AdvHotpotQA and 65.67 vs 54.1 on QALD-10-en (both vs ToG-2)—which we attribute to the combination of plan-aligned exploration, relation-wise clustering with deferred decompression, and a learned completeness judge that halts expansion once sufficient evidence is assembled. For fact verification, GRAPE is essentially on par with ToG on CREAK (92.77 vs 93.8) but substantially better on FEVER (79.12 vs 52.7 for ToG and 58.5 for CoK). We note that CREAK is a binary (true/false) setting that can often be resolved with shallow Wikidata evidence—both systems readily reach the first-level connections and, using a similar reasoning LLM, end up close. FEVER, in contrast, is a three-way decision where many ToG errors fall into the Not Enough Information class; GRAPE naturally mitigates this by passing multiple candidate paths to the final reasoning step when completeness has not yet marked a query as solved. On the remaining datasets GRAPE is broadly on par in accuracy; crucially, as shown next, it achieves this while considerably reducing execution time and the number of LLM calls.

## 4.4 INFERENCE COST

A key practical aspect is inference cost. By design, GRAPE keeps LLM usage small: in the worst case it makes $D_{\max} + 2$ calls—at most one per depth level for seed disambiguation, plus one for planning and one for final reasoning—independent of the beam width $b$. This is well below agent-style methods whose calls grow with both $b$ and $D_{\max}$, e.g., $2bD_{\max} + D_{\max} + 1$ for ToG and approximately $3D_{\max}$ for ToG-2. Empirically (Table 2), GRAPE reduces average LLM calls by 85.3% versus ToG (2.4 vs. 16.3), 55.6% versus ToG-2 (2.4 vs. 5.4), 69.2% versus PoG (2.4 vs. 7.8), and 78.2% versus CoK (2.4 vs. 11.0). The same trend holds for latency: average execution time per query drops by 85.0% versus ToG (10.4s vs. 69.3s), 61.9% versus ToG-2 (10.4s vs. 27.3s), 69.9% versus PoG (10.4s vs. 34.6s), and 65.4% versus CoK (10.4s vs. 30.1s). These savings stem from the encoder-driven exploration and the completeness judge controlling overexpansion.

## 4.5 ABLATION STUDY ON EXPLORATION COMPONENTS

We analyze the exploration stack by toggling four modules while keeping the rest of the pipeline fixed (planner/final–answer LLM, Wikidata KG, $b$=10, $D_{\max}$=3). *Expansion* is always present and is not ablated.

**Components.**

1. *Bi-encoder retriever.* Shortlists candidate continuations $P' = P \oplus (e_m, r, v)$ by semantic similarity (query/path text) before fine scoring.

2. *Cross-encoder reranker.* Computes $s^{\mathrm{rank}}(T_{X_k}, q, (e_m, r, v) \mid P)$ by comparing the incident triplets $T_{X_k}$ to the candidate $P'$.

3. *Completeness classifier.* Predicts $h_\theta(T_{X_k}, P') \in \{\text{STOP}, \text{CONTINUE}, \text{DROP}\}$ to control halt/expand/drop decisions.

4. *Relation-wise clustering (deferred decompression).* Groups $\mathcal{N}(e_m)$ by relation; selects a representative per relation and only decompresses a cluster if selected.

**Ablation protocol (substitutions when a module is off).** If the *bi-encoder* is off, we prefilter with BM25. If the *cross-encoder* is off, the LLM returns a scalar score for $(T_{X_k}, P')$ used for ranking. If the *completeness classifier* is off, the LLM outputs one of $\{\text{STOP}, \text{CONTINUE}, \text{DROP}\}$. If *clustering*

| Variant | Bi-enc. retr. | Cross-enc. rerank | $h_\theta$ | Rel. clustering | Hits@1/Acc (%) | LLM calls (avg) | Time / query (s) |
|---|---|---|---|---|---|---|---|
| Baseline (LLM rank + LLM comp.) | N | N | N | N | 70.2 | 55.0 | 120.9 |
| V1 | Y | N | N | N | 73.0 | 50.5 | 90.2 |
| V2 | Y | Y | N | N | 78.5 | 14.8 | 38.6 |
| V3 | Y | Y | Y | N | 80.6 | 3.2 | 16.5 |
| V4 (full) | Y | Y | Y | Y | 81.8 | 2.4 | 10.4 |

Table 3: Ablation over exploration components. Baseline: plain expansion with LLM-based ranking and completeness (no encoders, no clustering). V1: + bi-encoder shortlist. V2: + cross-encoder $s^{\mathrm{rank}}$ (LLM no longer ranks). V3: + $h_\theta$ (exploration decisions no longer call the LLM; only plan/answer/seed tie-breakers remain). V4: + relation-wise clustering (full system).

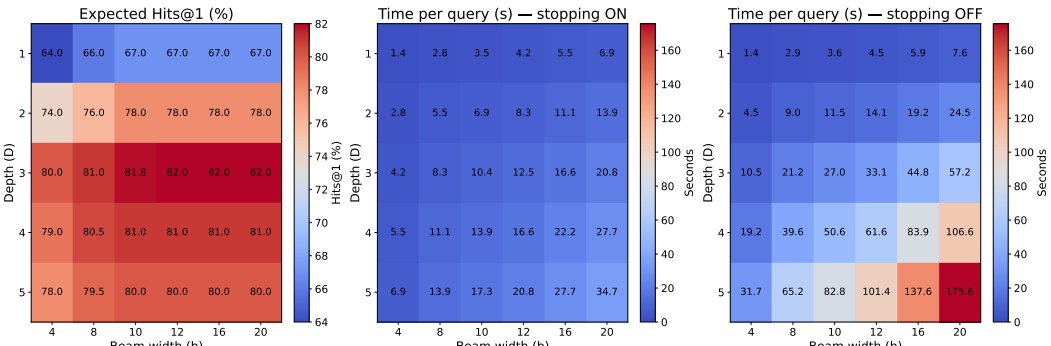

Figure 2: Beam–depth sweep. **Left:** Expected Hits@1 (%). **Middle:** Time/query (s) with $h_\theta$ ON. **Right:** Time/query (s) with $h_\theta$ OFF (shared color scale with the middle panel). Accuracy saturates around $b \approx 10$ and $D \approx 3$; without $h_\theta$, runtime grows sharply with depth due to the absence of early stopping and pruning.

is off, we expand all neighbors (optionally asking the LLM to choose a small set of relation types first).

As we can appreciate in Table 3, the LLM-native baseline is costly because the LLM both ranks many neighbors and decides completeness at each hop. Adding the bi-encoder (V1) reduces fan-out and latency but accuracy remains limited because ranking is still LLM-based. The cross-encoder (V2) is the inflection point: ranking moves off the LLM, average calls drop sharply and accuracy rises as promising candidates are promoted. Introducing $h_\theta$ (V3) removes LLM calls from exploration decisions, cutting calls to roughly three per query and further reducing time. Finally, relation-wise clustering (V4) eliminates redundant scoring on high–fan-out relations, yielding the lowest latency (10.4 s) with 2.4 average LLM calls.

### 4.6 WIDTH AND DEPTH VARIATIONS IN BEAM SEARCH

We study how beam width and maximum depth affect performance and inference time during exploration. We sweep beam widths $b \in \{4, 8, 10, 12, 16, 20\}$ and depths $D \in \{1, 2, 3, 4, 5\}$, keeping the rest of the pipeline fixed (Wikidata KG; planner/final–answer LLM; encoder stack). For each $(b, D)$ we report the average Hits@1 and the average execution time per query. We also report the time when the completeness module $h_\theta$ is deactivated (no early stopping / path dropping).

As shown in Figure 2, widening the beam helps up to $b \approx 10$ and then plateaus; increasing depth beyond $D \approx 3$ yields little gain and can slightly degrade accuracy (due to longer, noisier paths). With $h_\theta$ active, time grows roughly with $b \cdot D$ (encoder work), consistent with linear scaling in width and depth. When $h_\theta$ is disabled, time increases dramatically with depth (and mildly with width), reflecting the lack of early stopping and aggressive pruning. This highlights $h_\theta$'s role in containing search and keeping latency low.

| Backbone | Avg. Hits@1 (QA) | Avg. Acc. (Fact) | LLM calls (avg.) | Time/query (s) |
|---|---|---|---|---|
| GPT-4o (baseline) | **82.6** | **85.9** | 2.4 | 10.4 |
| GPT-3.5 | 79.5 | 83.0 | 2.4 | 9.0 |
| Llama-3 70B Instruct | 81.0 | 85.0 | 2.4 | 11.6 |
| Mistral Large Instruct | 80.0 | 84.0 | 2.4 | 11.0 |
| Claude 3.5 Sonnet | 81.8 | 85.5 | 2.4 | 11.2 |

Table 4: Backbone comparison with GRAPE ($b$=10, $D$=3). QA average aggregates WebQSP, AdvHotpotQA, QALD-10-en, Zero-Shot RE, CWQ, SimpleQuestions, WebQuestions, and T-REx; Fact average aggregates FEVER and CREAK. Comparison with GPT-3.5 OpenAI (2023), Llama-3 70B Instruct AI (2024a), Mistral Large Instruct AI (2024b), and Claude 3.5 Sonnet Anthropic (2024).

### 4.7 BACKBONE SENSITIVITY

Our main GRAPE results (Hits@1) use `GPT-4o` as the reasoning backbone. To assess backbone sensitivity, we swap only the planner/final–answer LLM for comparably capable models, while keeping the encoder stack (bi-/cross-encoder), $h_\theta$, relation clustering, and Wikidata fixed ($b$=10, $D$=3). We report averages over QA tasks (Hits@1) and fact checking (Acc.), together with average LLM calls and time per query. Because exploration is encoder-driven, LLM calls remain essentially unchanged across backbones; latency varies modestly with each model's generation speed.

GRAPE's performance is largely robust to the backbone because retrieval, path scoring, and stopping are encoder-driven. `GPT-4o` yields the strongest QA average (82.6) and fact accuracy (85.9); smaller or instruction-tuned alternatives trail by 1–3 pp on QA and 1–3 pp on fact checking. Average LLM calls remain $\approx$ 2.4 across backbones, while latency shifts modestly with model speed (e.g., slightly faster with `4o-mini`, slightly slower for larger open models). Overall, backbone choice nudges final precision, but GRAPE's efficiency and most of its accuracy gains come from the exploration stack rather than the reasoning LLM.

## 5 LIMITATIONS AND FUTURE WORK

GRAPE is a KGQA–centric system: unlike hybrid RAG approaches (e.g., ToG-2), integrating non-graph evidence (free text, tables, web) is non-trivial and would require additional alignment modules. The system is also sensitive to entity linking errors and KG coverage/staleness, which can yield premature stops or spurious paths, and to distribution shift in the trained components (bi-/cross-encoders and $h_\theta$), which may necessitate re-tuning. Finally, answer verbalization remains LLM-based and can introduce occasional aggregation or normalization mistakes. At present, the order of anonymous-entity disambiguation is seeded by the LLM's initial plan; a promising direction is to combine this with graph-derived statistics (e.g., node uncertainty or expected information gain) to prioritize easier decisions and prune earlier, thereby reducing end-to-end latency.

## 6 CONCLUSION

In this paper we introduced GRAPE, a KGQA framework that shifts retrieval from LLM-directed search to encoder-guided fuzzy graph matching with explicit stopping. In evaluation, GRAPE reduces both LLM calls and end-to-end latency by 60–80%, improves accuracy on 6 of 10 datasets, and remains competitive on the remainder. Our key insight is that effective multi-hop reasoning over KGs requires LLMs for plan induction and answer synthesis—not for steering the search. Consequently, this encoder-driven design offers a practical path to faster, more scalable KGQA without sacrificing accuracy.

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

## A  ADDITIONAL IMPLEMENTATION DETAILS

### A.1  PROMPTS

Prompt for generating the anonymous graph:

```
We need to make triplets to make a query from a given text, where the
    unknown entities or verbs to match are represented by an X_i.
The idea is to subtitle an entity that is unkwown by X_i, this entities
    might be parsed later using a triplet extracted from a knowledge
    graph.
Try to use relations as verbs and entities as nouns, although is not
    mandatory if the relations are complex.
Take as anchor entity the one that is more restrictive, i.e. the one that
     is more specific.
There should always be at least X_1 but there might be more than one
    entity to match, so be as fine-grained creating entities as possible,
     the more the better and careful with modifiers like dates or numbers
Note that we should consider every sentence as positive as our goal is
    just to generate a query for checking information. Therefore ignore
    negations and quantifiers like never, not, only, exclusivy, always
    ...
Here are a few examples:
```

– "John Wick is a movie starting the american actor Keenau Reeves and
directed by Stephen Spielberg" -> [(John Wick, starring, X_1), (X_1,
country_of_origin, United States), (John Wick, directed_by, X_2)].
– "Barack Obama is married with a lady who has a kid that has blue eyes"
-> [(Barack Obama, married, X1), (X1, has_children, X2), (X2,
has_eye_color, blue)].
– "Berlin is the capital city of Germany" -> [(Berlin, is_capital_of,
Germany)].
– "Paul MacCartney has never been in a band" -> [(X_1, is, band), (X_1,
is_member, Paul MacCartney)].
– "John Doe had dinner with Barack Obama" -> [(John Doe, X_i, Barack
Obama)].
– "Duke Leto Attreides is a character in the Dune poem" -> [(Duke Leto
Attreides, is_character_in, X_1), (X_1, is, poem)].
– "The movie Titanic was directed by a Canadian man." -> [(X_1,
director_of, Titanic), (X_1, is, man)].
– "The movie Titanic starred an actress." -> [(X_1, starring, Titanic), (
X_1, is, actress)].
– "Don McLean starred with a British actress and a German actor in a 2016
movie." -> [(Don McLean, starring, X_1), (X_1, release_date, 2016),
(X_2, starring, X1), (X_2, nationality, British), (X_2, profession,
actress), (X_3 starring, X_1), (X_3, profession, actor). (X_3,
nationality, German)].
– "The movie Breakfast at Tiffany's featured a song that was not composed
by Henry Mancini." -> [(Breakfast at Tiffany's, featured, X_1), (X_1
, is, song), (X_1, composed_by, X_2), (X_2, is_not, Henry Mancini)].
– "Which TV show Barack Obama appeared in?" -> [(Barack Obama,
appeared_in, X_1), (X_1, is, TV show)].
– "The actor who played Jack Ryan in the movie The Hunt for Red October
(1969) is from the same country as Barack Obama" -> [(X_1,
interpreted, Jack_Ryan), (Jack_Ryan, character_of,
The_Hunt_of_the_Red_October), (The_Hunt_Of_The_Red_October,
release_date, 1969), (X_1, nationality, X_2), (Barack_Obama,
nationality, X_2)].
– "The director of the movie Move (2010) is from the same country as the
director of the movie The Dark Knight" -> [(X_1, director_of, Move),
(Move, release_date, 2010), (X_1, nationality, X_2), (X_3,
director_of, The_Dark_Knight), (X_3, nationality, X_2)].
– "The film The Dark Knight was released in the same year as the film The
Hunt for Red October" -> [(The_Dark_Knight, release_date, X_1), (
The_Hunt_Of_The_Red_October, release_date, X_1)].
– "Who is the paternal grandfather of John Doe?" -> [(X_1, father_of,
John_Doe), (X2, father_of, X_1)]. (Note here the grandfather relation
can be splitted in two)
– "Who is the maternal grandfather of John Doe?" -> [(X_1, mother_of,
John_Doe), (X2, father_of, X_1)].
– "From which universities are the Canadian recipients of the Turing
Award that work in the field of Deep Learning?" -> [(X_1, nationality
, Canadian), (X_1, received, Turing_Award), (X_1, field_of_work,
Deep_Learning), (X_1, studied_at, X_2), (X_2, is, university)].

Parenthesis near known entities usually give some information about the
entity, like the year of a movie, profession of some person, a
clarification of what the entity is, or place of an event, like The
Hunt for Red October (1969) -> [(The_Hunt_Of_The_Red_October,
release_date, 1969)].
Note that for example in the last ones even if we matched X_1 we would
still need to check that X_1 is a TV_show.
However, it's also extremly important not to create dummy entities like (
X_1, name, Duke Leto Attreides) or (X_1, is, Duke Leto Attreides) as
this is not useful for querying.

Return a list (i.e [triplet, triplet, triplet ....]) of triplets
representing the following text, keep it short and simple but be as

```
    specific as possible in the entities, and *be careful not to drop
    information about the X_i entities*
```

Prompt for reasoning:

```
You are given:
1. A query expressed as triplets with anonymous entities (X_i).
2. The target entity to resolve (e.g., X_1).
3. A list of candidate answers for the target entity.
4. A set of logical paths extracted from a knowledge graph.

Your task:
- Identify which candidate answer best resolves the target entity.
- Replace the anonymous entities (X_i) in the query triplets with the
    resolved entities from the logical path.
- If no candidate matches, answer "None".

---

Example:

Target entity: X_1

Query triplets:
[(X_1, director_of, Titanic), (X_1, nationality, Canadian), (Titanic,
    release_date, 1997)]

Candidates:
- James Cameron
- Steven Spielberg
- Denis Villeneuve

Logical paths:
- James Cameron director_of Titanic     James Cameron nationality
    Canadian     Titanic release_date 1997
- Steven Spielberg director_of Titanic     Steven Spielberg nationality
    American     Titanic release_date 1997
- Denis Villeneuve director_of Arrival     Denis Villeneuve nationality
    Canadian     Arrival release_date 2016

Answer:
James Cameron
```

A.2 TRAINING THE ANONYMOUS-GRAPH ENCODER

**Goal.** We train an encoder to score the compatibility between an *anonymous query subgraph* and a *candidate KG path/subgraph*, so that fuzzy matches (synonyms, variable-length alignments, type-consistent substitutions) receive high scores while incomplete or corrupted matches receive low scores.

**Supervision source.** From the train/dev splits of all KGQA benchmarks, we convert each query $q$ into its AKG $\hat{\mathcal{G}}_q = (E_q, R_q, T_q)$ and identify the subset $T_X \subseteq T_q$ of triplets incident to a chosen anonymous variable $X$ (the variable to resolve at this step). When gold supporting paths are available, we use them; otherwise, we mine minimal satisfying paths in the KG that instantiate $X$ and make the query true.

**Positive examples.** Starting from $(T_X, P^\star)$ pairs (where $P^\star$ is a supporting path), we create *hard positives* via: (i) **synonymic rewrites** of relations/entities (from a curated synonym list) that preserve truth conditions; (ii) **neutral triplet insertion**—add edges touching nodes in $T_X$ but not changing the answer set (e.g., types/attributes); (iii) **query refinements**—add logically implied constraints (e.g., *instance_of*(X, *Film*)). All such augmentations keep the label positive.

---

**Algorithm 1** Anonymous-Graph Training Pair Generation

---

**Require:** Datasets $\mathcal{D}$, KG $\mathcal{G}$, synonym maps $\mathsf{SynRel}, \mathsf{SynEnt}$, topology patterns $\mathsf{Topo}$, augmentation budget $K$

**Ensure:** Training set $\mathcal{S} = \{(T_X, P, y)\}$

1: $\mathcal{S} \leftarrow \varnothing$
2: **for all** $(q, \text{answers}) \in \mathcal{D}_{\text{train}} \cup \mathcal{D}_{\text{dev}}$ **do**
3: $\quad \hat{\mathcal{G}}_q = (E_q, R_q, T_q) \leftarrow \text{AnonGraph}(q)$
4: $\quad$ **for all** $X \in \text{AnonymousVars}(\hat{\mathcal{G}}_q)$ **do**
5: $\qquad T_X \leftarrow \{(h, r, t) \in T_q \mid h = X \vee t = X\}$ $\qquad\qquad\qquad$ ▷ constraints incident to $X$
6: $\qquad \mathcal{P}^\star \leftarrow \text{MineSupportingPaths}(T_X, \mathcal{G})$ $\qquad\qquad\qquad$ ▷ use gold if available
7: $\qquad$ **for all** $P^\star \in \mathcal{P}^\star$ **do**
8: $\qquad\quad \mathcal{S} \leftarrow \mathcal{S} \cup \{(T_X, P^\star, 1)\}$ $\qquad\qquad\qquad\qquad\qquad$ ▷ base positive
9: $\qquad\quad$ **for** $k = 1$ to $K$ **do** $\qquad\qquad\qquad\qquad\qquad\qquad$ ▷ augment positives
10: $\qquad\qquad T_X^+ \leftarrow \text{SynonymicRewrite}(T_X, \mathsf{SynRel}, \mathsf{SynEnt})$
11: $\qquad\qquad T_X^+ \leftarrow \text{InsertNeutralTriplets}(T_X^+, \mathcal{G})$
12: $\qquad\qquad T_X^+ \leftarrow \text{AddImpliedConstraints}(T_X^+)$
13: $\qquad\qquad P^+ \leftarrow \text{TopologyJitter}(P^\star, \mathsf{Topo})$ $\qquad\qquad\qquad$ ▷ equivalent detours
14: $\qquad\qquad \mathcal{S} \leftarrow \mathcal{S} \cup \{(T_X^+, P^+, 1)\}$
15: $\qquad\quad$ **end for**
16: $\qquad\quad$ **for** $k = 1$ to $K$ **do** $\qquad\qquad\qquad\qquad\qquad\qquad$ ▷ generate hard negatives
17: $\qquad\qquad T_X^-, P^- \leftarrow \text{CorruptOne}\big(T_X, P^\star; \text{drop-edge} \vee \text{bad-syn} \vee \text{target-swap} \vee$ topology-perturb$\big)$
18: $\qquad\qquad$ **if** $\neg\text{Satisfies}(T_X^-, P^-)$ **then**
19: $\qquad\qquad\quad \mathcal{S} \leftarrow \mathcal{S} \cup \{(T_X^-, P^-, 0)\}$
20: $\qquad\qquad$ **end if**
21: $\qquad\quad$ **end for**
22: $\qquad$ **end for**
23: $\quad$ **end for**
24: **end for**
25: **return** $\mathcal{S}$

---

**Negative examples.** We generate *hard negatives* by minimally breaking support: (i) **edge dropout**—remove a key hop in $P^\star$ or a required constraint in $T_X$; (ii) **relation/entity corruption**—replace a relation or entity with a near-synonym that changes truth conditions; (iii) **target corruption**—swap the answer with a type-consistent non-answer; (iv) **topology perturbations**—reroute or re-order hops to produce misleading chains. These produce pairs $(T_X, \tilde{P})$ that *look* similar but do not satisfy $T_X$.

**Order-invariance (sets, not sequences).** To learn over *sets of triplets*, we randomly permute the triplets in $T_X$ at sampling time and re-linearize them (and optionally shuffle non-critical hops in the candidate) so the encoder does not overfit to serialization order.

**Input formatting and objective.** We serialize $(T_X,\ P)$ as two text fields (or graph tokens) and feed them to a cross-encoder to obtain a score $s(T_X, P) \in \mathbb{R}$. We train with binary labels $y \in \{0, 1\}$ using a logistic loss:

$$\mathcal{L} = -y \log \sigma\big(s(T_X, P)\big) - (1 - y) \log\big(1 - \sigma(s(T_X, P))\big),$$

or, optionally, a pairwise margin loss for ranking positives above negatives sampled for the same $T_X$.

**Topology diversity.** To avoid overfitting to chains, we synthesize examples spanning chains, forks, diamonds, and short detours (1–3 hops), ensuring the encoder sees multiple structures that can satisfy the same $T_X$.

---

**Algorithm 2** Batch sampling with set invariance

---

**Require:** Training set $\mathcal{S}$, permutations per item $M$
1: **for** each minibatch **do**
2:      Sample $(T_X, P, y)$ items from $\mathcal{S}$
3:      **for all** items in batch **do**
4:          $T_X^{(\pi)} \leftarrow \text{RandomPermutation}(T_X)$
5:          $P^{(\pi)} \leftarrow \text{OptionalShuffle}(P)$                         ▷ shuffle non-critical hops
6:          Encode pair $\left(T_X^{(\pi)}, P^{(\pi)}\right)$ and update via logistic (or margin) loss
7:      **end for**
8: **end for**

---

## A.3 HARDWARE SPECIFICATIONS

All experiments were performed in a machine with the technical capabilities reported in section A.3.

| CPU | AMD Ryzen Threadripper 3975WX |
|-----|-------------------------------|
| RAM | 256 GB |
| Cores | 64 |
| GPU | 2x Nvidia A100 160GB |

Table 5: Specifications of the machine in which the experiments were executed.

## A.4 TRAINING HYPERPARAMETERS

The full table of hyperparameters used in the training of the path cross encoder systems and inference for LLMs can be found in table 6. Different options for the setting tried of the system appear between curly braces, while the selected ones appear in bold.

| Parameter | Value |
|-----------|-------|
| Optimizer | AdamW |
| Learning Rate | $\{10^{-7}, 10^{-6}, \mathbf{10^{-5}}, 10^{-4}\}$ |
| Gradient Accumulation Steps | $\{1, \mathbf{5}, 10\}$ |
| Maximum Gradient Norm | $\{\mathbf{1}, 5, 10, 50, 100\}$ |
| Batch Size | $\{4, 16, 32, \mathbf{64}, 128\}$ |
| Epochs | 1, 5, **10**, 15, 20 |
| Evaluation Steps | 1000 |
| Scheduler | $\{\mathbf{\text{Cosine Annealing}}, \text{Linear}\}$ |
| Weight Decay | 0.01 |
| Maximum Gradient Norm | **1**, 5, 10 |
| Loss Function | Cross-Entropy with logits |
| Max Tokens | 512 |
| $D_max$ (maximum expansion depth) | 3 |
| $b$ (beam size) | 10 |
| top_p | 1 |
| temperature | 0.0 |
| max_generated_tokens | 2048 |

Table 6: Training hyperparameters for the proposed system. Between curly braces are all values tested during optimization, the one selected are marked in bold.

## A.5 KG EXPLORATION PSEUDOCODE

---

**Algorithm 3** Anonymous KG Exploration with Relation Clustering

---

**Require:** Query $q$; initial $\widehat{G}^0$; beam size $b=10$; depth limit $D_{\max}$
1: Initialize frontier $\mathcal{F}^0 \leftarrow \{\emptyset\}$, accepted set $\mathcal{A} \leftarrow \varnothing$, priority queue $\mathsf{Q} \leftarrow \varnothing$
2: **for** $k = 0, 1, 2, \ldots$ **do**
3:     **for each** $P \in \mathcal{F}^k$ **do**
4:         Select anonymous target $\hat{x} \in \widehat{E}$ referenced by the next planned triplet $\tau^\star = (\hat{x}, r^\star, o^\star)$
5:         Let $e$ be the current binding for $\hat{x}$ reached along $P$
6:         Enumerate $\mathcal{N}(e) = \{(r, v)\}$ and cluster: $\mathcal{N}(e) = \bigsqcup_r \mathcal{C}_r$
7:         **for each** cluster $\mathcal{C}_r$ **do**
8:             $v_r^\star \leftarrow \arg\max_{(r,v)\in\mathcal{C}_r} s_{\mathrm{rank}}\big(q, (e, r, v) \mid P\big)$
9:         **end for**
10:        $\mathcal{R} \leftarrow \{(e, r, v_r^\star)\}_r$
11:        Candidate continuations $\mathcal{C} \leftarrow \{P \oplus (e, r, v_r^\star) : (e, r, v_r^\star) \in \mathcal{R}\}$
12:        $\widetilde{\mathcal{F}} \leftarrow \mathrm{TOPB}(\mathcal{C}, b \text{ by } s_{\mathrm{rank}})$
13:        **for each** $P' \in \widetilde{\mathcal{F}}$ **do**
14:            $y \leftarrow h_\theta(q, P') \in \{\text{COMPLETE}, \text{EXPAND}, \text{IRRELEVANT}\}$
15:            **if** $y = \text{IRRELEVANT}$ **then**
16:                **continue**                                  $\triangleright$ drop
17:            **else if** $y = \text{COMPLETE}$ **or** $\mathrm{depth}(P') = D_{\max}$ **then**
18:                $\mathcal{A} \leftarrow \mathcal{A} \cup \{P'\}$              $\triangleright$ accept and stop expanding
19:            **else**
20:                Push $P'$ into $\mathsf{Q}$ with key $u(P')$          $\triangleright$ mark for expansion
21:            **end if**
22:        **end for**
23:     **end for**
24:     $\mathcal{F}^{k+1} \leftarrow \varnothing$
25:     **while** $|\mathcal{F}^{k+1}| < b$ **and** $\mathsf{Q} \neq \varnothing$ **do**
26:         $P^\star \leftarrow \mathrm{POPMAX}(\mathsf{Q})$
27:         **if** $P^\star$ chose representative $(e, r, v_r^\star)$ for expansion **then**
28:         *// cluster decompression on demand*
29:             Insert $P^\star \oplus (e, r, v)$ for top-$k$ members $(r, v) \in \mathcal{C}_r$ by $s_{\mathrm{rank}}$
30:         $\mathcal{F}^{k+1} \leftarrow \mathcal{F}^{k+1} \cup \{P^\star\}$         $\triangleright$ advance beam; enables backtracking if needed
31:     **end while**
32:     **if** $\mathsf{Q} = \varnothing$ **then**
33:         **break**
34:     **end if**
35: **end for**
36: **return** $\mathcal{A}$

---

