# OpenReview forum: "GRAPE: Graph Reasoning with Anonymous Path Encoders"
_ICLR.cc/2026/Conference — ICLR 2026 Conference Withdrawn Submission_

### Official Review · Reviewer_TFcT · 2025-10-27

**Soundness:** 2
**Presentation:** 2
**Contribution:** 2
**Rating:** 2
**Confidence:** 4

**Summary:**

The paper proposes GRAPE, a knowledge-graph-centric alternative to LLM-agent style KGQA systems. Instead of driving every traversal step through an LLM, GRAPE reformulates reasoning as fuzzy anonymous graph matching: questions are parsed into an Anonymous Knowledge Graph (AKG), then resolved via encoder-only retrieval and completeness classification, with only one final LLM call for aggregation. By offloading traversal entirely to encoders, GRAPE achieves up to 85% faster inference while matching or beating SOTA accuracy on ten KGQA, fact-checking, and open-domain QA datasets.

**Strengths:**

1. It proposes a novel framework that relieves LLMs from the burden of multi-step graph traversal in KGQA tasks with a lightweight path retrieval and classification module.
2. This approach significantly improves inference efficiency while maintaining or improving accuracy compared to SOTA LLM-agent methods.

**Weaknesses:**

1. Limited novelty: The proposed framework shares similarities with existing retrieval-augmented methods in KGQA by developing a efficient ranking module to select relevant paths from knowledge graphs for expansion.
2. Limited generalizability: As the path retrieval module is trained specifically for datasets, it may not generalize well to unseen knowledge graphs or domains without retraining or fine-tuning.
3. Lack of strong baselines: The experiments mainly compare against LLM-agent methods, but do not include strong training-based methods like GNN-based approaches [1,2]. Thus it is unclear how GRAPE compares to these alternatives in terms of accuracy and efficiency.
4. Lack of ablation studies: The paper does not include ablation studies to analyze the contributions of different components, especially the path retrieval and classification modules. Can we achieve similar performance with some existing retrieval methods like BM25 or dense retrieval like Qwen3-emb/ranker?
5. Lack of proper metrics: The paper focuses on Hits@1 but does not report other important metrics like F1, which are commonly used in KGQA to evaluate overall answer coverage.
6. Mis-citations / Incomplete Citations: AdvHotpotQA seems to be mis-cited. RRKG citation is missing. Citations of line 088 is missing.

[1] Mavromatis, Costas, and George Karypis. "Gnn-rag: Graph neural retrieval for efficient large language model reasoning on knowledge graphs." Findings of the Association for Computational Linguistics: ACL 2025. 2025.

[2] Li, Mufei, Siqi Miao, and Pan Li. "Simple is Effective: The Roles of Graphs and Large Language Models in Knowledge-Graph-Based Retrieval-Augmented Generation." The Thirteenth International Conference on Learning Representations.

**Questions:**

Please see my weaknesses 1-5.

---

### Official Review · Reviewer_awUu · 2025-10-28

**Soundness:** 2
**Presentation:** 2
**Contribution:** 2
**Rating:** 4
**Confidence:** 4

**Summary:**

This paper introduces GRAPE (Graph Reasoning with Anonymous Path Encoders), a framework that leverages path encodings over uncertain nodes and relations in knowledge graphs (KGs) to heuristically guide KB navigation. Rather than depending on a fully LLM-native retrieval pipeline, GRAPE replaces repeated model calls with encoder-only models that act as a semantic fuzzy query-matching engine. Experiments across multiple multi-hop KGQA benchmarks show that GRAPE achieves substantial speedup compared with LLM-based pipelines while achieving state-of-the-art performance.

**Strengths:**

**S1.** The paper is overall easy to follow.

**S2.** The paper experiments with a large number of datasets.

**S3.** Empirical studies report standard deviations across five runs.

**S4.** The paper performs ablation studies for various components.

**S5.** The paper performs efficiency evaluation in terms of LLM calls and wall clock time.

**Weaknesses:**

**W1.** The paper fails to cite, discuss and / or compare against (verbally or empirically) many highly relevant and impactful works, which is crucial for evaluating the paper's contributions, e.g., novelty of the proposed approach.

- Efficient encoder-based KG retrieval without LLM-based explorations ([1], [2], [3]), which aligns with the idea of encoder-based efficient graph exploration of the proposed approach.

- KG-based RAG with semantic parsing, which generates templates for relation path matching similar to the idea of Anonymous Knowledge Graph Matching. E.g., [4] fine-tunes an LLM to predict anonymized relation paths for matching in the KG, which is very similar to the idea of GRAPE. [5] also employs LLMs like ChatGPT to generate SQL queries for searching over the KG.

- Other related RAG approaches: [6], [7], [8]

[1] He et al. G-Retriever: Retrieval-Augmented Generation for Textual Graph Understanding and Question Answering. NeurIPS 2024.

[2] Mavromatis et al. GNN-RAG: Graph Neural Retrieval for Efficient Large Language Model Reasoning on Knowledge Graphs. ACL Findings 2025.

[3] Li et al. Simple Is Effective: The Roles of Graphs and Large Language Models in Knowledge-Graph-Based Retrieval-Augmented Generation. ICLR 2025.

[4] Luo et al. Reasoning on Graphs: Faithful and Interpretable Large Language Model Reasoning. ICLR 2024.

[5] Li et al. Chain-of-knowledge: Grounding large language models via dynamic knowledge adapting over heterogeneous sources. ICLR 2023.

[6] Chen et al. Plan-on-Graph: Self-Correcting Adaptive Planning of Large Language Model on Knowledge Graphs. NeurIPS 2024.

[7] Luo et al. Graph-constrained Reasoning: Faithful Reasoning on Knowledge Graphs with Large Language Models. ICML 2025.

[8] Chen et al. PathRAG: Pruning Graph-based Retrieval Augmented Generation with Relational Paths. arXiv 2025.

**W2.** The methodology (Section 3.2) discussion assumes that all queries can be represented as anonymous chain graphs while in practice multi-hop queries can have parallel branching substructures, leading to general directed acyclic graphs. How to order sub-queries / anonymous variables in such cases is not discussed at all.

**W3.** The cluster representative selection mechanism can be inferior for sub-queries with multiple valid answers. E.g., which books were authored by XXX.

**W4.** The empirical studies do not consider the F1 metric, which is also widely adopted for KGQA and KG-based RAG evaluation that reflects precision and recall.

**W5.** Minor presentation issues.
- The format of in-text citations needs to be corrected throughout the paper with proper use of ~\citep{} and ~\citet{}. E.g., Petroni et al. (2021) should be (Petroni et al., 2021).
- There are in-text citations not properly rendered, displayed as "?" in the text. E.g., "retrieval pipelines require repeated LLM calls Chen et al. (2023); Oche et al. (2025); Wang et al. (2024); ?."

**Questions:**

**Q1.** How do you combine the scores obtained by the retriever and reranker? The introduction in Section 3.3 is not clear enough.

**Q2.** Why did you not also experiment with alternative KGs like Freebase?

**Q3.** Why not report the efficiency study of RoG in Table 2?

---

### Official Review · Reviewer_TwUU · 2025-10-29

**Soundness:** 3
**Presentation:** 3
**Contribution:** 3
**Rating:** 2
**Confidence:** 4

**Summary:**

For knowledge graph question answering tasks, a method is proposed that uses embeddings to match entities sequentially to find paths, which is faster than existing methods that repeatedly execute using LLM.

**Strengths:**

S1. The method proposed in the paper indeed shows good consideration for efficiency, and effectively improving efficiency is a significant advancement.

**Weaknesses:**

W1. Limited Methodological Novelty and High Complexity. The proposed method's workflow is highly complex and consists largely of orchestrating existing, pre-trained models, including embedding models and LLMs. This limits its methodological novelty. In fact, prior to the advent of LLMs, many approaches in the Knowledge Graph Question Answering (KGQA) domain employed key steps that are conceptually similar to this method. The proposed approach appears to be more of a hybrid combination of pure LLM-based reasoning and pure embedding-based matching, rather than a fundamentally new framework.

W2. Gaps in Experimental Performance. In terms of experimental results, the proposed method still demonstrates a noticeable performance gap when compared to the current state-of-the-art on several mainstream benchmarks, such as WebQSP, CWQ, and Simple Questions. While it achieves commendable results on some other datasets, its performance on these key, widely-used benchmarks indicates that it has not yet surpassed the leading methods in the field.

W3. Potentially Unfair Experimental Configuration. The paper's use of GPT-4o as the foundation model may create an unfair comparison with existing work. Many prior LLM-dependent methods were benchmarked using earlier, less capable models. This discrepancy makes it difficult to assess the true contribution of the proposed methodology, as a portion of the performance gains may be attributable to the superior capabilities of the more advanced base model rather than the novelty of the method's design itself.

**Questions:**

Q1. Justification for Knowledge Graph-based Answering. The paper needs to better justify the necessity of using a Knowledge Graph (KG) for question answering. Could the questions not be answered directly by a contemporary LLM augmented with web retrieval? It is well-known that many existing KGs contain a significant number of errors. In contrast, modern LLMs already possess vast internal knowledge, and when combined with real-time web search, they may achieve performance that exceeds that of an LLM relying on a potentially flawed KG. I request that the authors respond to this point, ideally by providing experimental results and specific case studies that demonstrate the advantages of the KG-based approach over a strong LLM+retrieval baseline.

Q2. Missing Equation Numbers. The equations presented in the manuscript lack numbering, which makes them difficult to reference and discuss. Please add equation numbers throughout the paper for clarity.

Q3. Flawed Definition of KGQA. The definition of KGQA provided in the equations on lines 153 and 154 appears to be problematic and incomplete. Specifically:
What is the meaning of the expression q \wedge e = e_k? This notation is unclear.
The term e_0 is used without a prior definition. Please clarify what it refers to.
Furthermore, the formulation seems to imply that KGQA is limited to querying for entities, while ignoring the querying of relations. This appears to be an overly narrow definition of the KGQA task. Please revise and complete this definition.

Q4. Limitations of the Anonymous Entity Solving Approach. While the authors propose the concept of "anonymous entity solving on a graph," this approach does not seem fundamentally different from existing methods where an LLM sequentially determines the entities along a path. More importantly, this method appears to be designed only for scenarios where the entities are uncertain but the relations are known. How would the framework handle the inverse situation, where the relationship itself is uncertain? For example, in a question like, "What is the relationship between Kobe Bryant and LeBron James?" the entities are known, but the relation is the object of the query. Additionally, how does the method handle queries that require reasoning over multi-hop relations?

---

### Official Review · Reviewer_XLQY · 2025-10-31

**Soundness:** 1
**Presentation:** 1
**Contribution:** 2
**Rating:** 2
**Confidence:** 5

**Summary:**

This paper introduces graph reasoning with anonymous path encoder (GRAPE) framework, which leverages path encodings over uncertain nodes and relations in the knowledge graphs to heuristically guide KB navigation. To be specific, GRAPE uses encoder-only models to act as a semantic fuzzy query-matching engine. Extensive experiments show the effectiveness of the proposed method.

**Strengths:**

1.	Integrating LLMs with KGs is a critical research area.
2.	Extensive experiments show the effectiveness of the proposed method.

**Weaknesses:**

1.	There may lack some representative recent baseline methods for comparison, such as GNN-RAG-RA [1], SubgraphRAG [2].
2.	It would be beneficial to further refine the figures and tables to enhance readability. For instance, the font size in Figure 1 is relatively small.
3.	For efficient analysis experiments, it would be beneficial to include retrieved-based methods (e.g., RoG) for comprehensive comparison.


[1] Mavromatis, Costas, and George Karypis. "Gnn-rag: Graph neural retrieval for large language model reasoning." arXiv preprint arXiv:2405.20139 (2024).

[2] Li, Mufei, Siqi Miao, and Pan Li. "Simple is Effective: The Roles of Graphs and Large Language Models in Knowledge-Graph-Based Retrieval-Augmented Generation." The Thirteenth International Conference on Learning Representations.

**Questions:**

Please refer to the **weaknesses** section.

---

### Note · Authors · 2025-11-12

I have read and agree with the venue's withdrawal policy on behalf of myself and my co-authors.